# Antibiotic Prescription and In-Hospital Mortality in COVID-19: A Prospective Multicentre Cohort Study

**DOI:** 10.3390/jpm12060877

**Published:** 2022-05-26

**Authors:** Larisa Pinte, Alexandr Ceasovschih, Cristian-Mihail Niculae, Laura Elena Stoichitoiu, Razvan Adrian Ionescu, Marius Ioan Balea, Roxana Carmen Cernat, Nicoleta Vlad, Vlad Padureanu, Adrian Purcarea, Camelia Badea, Adriana Hristea, Laurenţiu Sorodoc, Cristian Baicus

**Affiliations:** 1Faculty of Medicine, Carol Davila University of Medicine and Pharmacy, 050474 Bucharest, Romania; cristian.niculae@drd.umfcd.ro (C.-M.N.); laura-elena.stoichitoiu@drd.umfcd.ro (L.E.S.); tane67@gmail.com (R.A.I.); cameliabadea72@yahoo.com (C.B.); adriana.hristea@umfcd.ro (A.H.); cristian.baicus@umfcd.ro (C.B.); 2Department of Internal Medicine, Colentina Clinical Hospital, 020125 Bucharest, Romania; 3Clinical Research Unit, Reseau d’Epidemiologie Clinique International Francophone, 020125 Bucharest, Romania; 4Department of Internal Medicine, Clinical Emergency Hospital Sfantul Spiridon, 700111 Iasi, Romania; alexandr.ceasovschih@yahoo.com (A.C.); laurentiu.sorodoc@gmail.com (L.S.); 5Faculty of Medicine, Grigore T. Popa University of Medicine and Pharmacy, 700115 Iasi, Romania; 6Department of Infectious Diseases, National Institute for Infectious Diseases Prof. Dr. Matei Bals, 021105 Bucharest, Romania; 7Department of Pneumology, Colentina Clinical Hospital, 020125 Bucharest, Romania; marius.balea@gmail.com; 8Faculty of Medicine, Ovidius University, 900527 Constanta, Romania; roxana.cernat@seanet.ro (R.C.C.); nicoleta.lalescu@yahoo.com (N.V.); 9Department of Infectious Diseases, Clinical Hospital of Infectious Diseases, 900178 Constanta, Romania; 10Department of Internal Medicine, University of Medicine and Pharmacy Craiova, 200349 Craiova, Romania; vlad.padureanu@umfcv.ro; 11Department of Internal Medicine, Craiova Emergency County Hospital, 200642 Craiova, Romania; 12Department of Internal Medicine, Sacele County Hospital, 505600 Brasov, Romania; adrian.purcarea@gmail.com

**Keywords:** antibiotics, antibacterial agents, COVID-19, SARS-CoV-2, mortality, hospital mortality, cohort studies, prospective studies

## Abstract

Background: Since the beginning of the COVID-19 pandemic, empiric antibiotics (ATBs) have been prescribed on a large scale in both in- and outpatients. We aimed to assess the impact of antibiotic treatment on the outcomes of hospitalised patients with moderate and severe coronavirus disease 2019 (COVID-19). Methods: We conducted a prospective multicentre cohort study in six clinical hospitals, between January 2021 and May 2021. Results: We included 553 hospitalised COVID-19 patients, of whom 58% (311/553) were prescribed antibiotics, while bacteriological tests were performed in 57% (178/311) of them. Death was the outcome in 48 patients—39 from the ATBs group and 9 from the non-ATBs group. The patients who received antibiotics during hospitalisation had a higher mortality (RR = 3.37, CI 95%: 1.7–6.8), and this association was stronger in the subgroup of patients without reasons for antimicrobial treatment (RR = 6.1, CI 95%: 1.9–19.1), while in the subgroup with reasons for antimicrobial therapy the association was not statistically significant (OR = 2.33, CI 95%: 0.76–7.17). After adjusting for the confounders, receiving antibiotics remained associated with a higher mortality only in the subgroup of patients without criteria for antibiotic prescription (OR = 10.3, CI 95%: 2–52). Conclusions: In our study, antibiotic treatment did not decrease the risk of death in the patients with mild and severe COVID-19, but was associated with a higher risk of death in the subgroup of patients without reasons for it.

## 1. Introduction

Since the beginning of the COVID-19 pandemic, empiric antibiotics (ATBs) have been prescribed on a large scale in both hospitalised patients and outpatients. In its early stages, almost three-quarters of coronavirus disease 2019 (COVID-19) patients received ATBs despite the low rates of confirmed bacterial infections—especially regarding pulmonary involvement [1,2]. Some guidelines reserved the use of antibiotics for strong clinical suspicion of bacterial infection [3]; others advocated for empiric administration in severe COVID-19 patients while assessing for de-escalation or reconsidering the indication daily [4,5].

However, since treatments used for COVID-19 can alter the classical inflammation markers, detecting a bacterial infection in the earlier phase of the pandemic was more difficult [6]. In addition, COVID-19 itself increases C-reactive protein (CRP) and procalcitonin values—markers for bacterial infection, whose values are associated with disease severity [7].

Moreover, colonisation is frequently confused with infection, and the isolation of a bacterium may automatically lead to antibiotic prescription in patients with severe disease [8].

Considering the increased risk of death in COVID-19 patients with bacterial infections, as well as the consequences of inappropriate antibiotic prescription, physicians faced the risk of overprescription.

The aim of our study was to assess the impact of antibiotic treatment on the outcomes of hospitalised patients with COVID-19, as well as document the types of bacteria isolated and antibiotics prescribed.

## 2. Materials and Methods

### 2.1. Study Design and Population

This was a prospective multicentre cohort study, conducted in 6 clinical hospitals in Romania. We included adult patients with confirmed COVID-19 admitted to the hospital between January 2021 and May 2021. Study participants were enrolled from departments of infectious diseases, internal medicine, and pneumology.

The inclusion criteria for the study were (a) patients 18 years of age or older, and (b) confirmed COVID-19 diagnosed by a positive real-time polymerase chain reaction (RT-PCR) test or SARS-CoV-2 rapid antigen test.

The exclusion criteria were (a) patients initially admitted to the ICU, (b) end-stage kidney disease undergoing haemodialysis or peritoneal dialysis, and (c) lympho- and myeloproliferative disorders.

The treatment decision remained at the discretion of the attending physician. We adhered to the Strengthening the Reporting of Observational Studies in Epidemiology reporting guidelines [9]. The flow diagram of the study is presented in Figure 1.

### 2.2. Ethical Consideration

This study was conducted in accordance with the principles of the Declaration of Helsinki, and accepted by the ethics committee of the medical centres involved (32/8 December 2020). Patients signed written informed consent during hospital admission and, due to biosecurity reasons, the consent was also collected verbally and registered in the medical chart by the treating physician.

### 2.3. Variables and Data Measurement

The uniformity of the cohort was ensured using the following criteria for classifying the form of the disease and the degree of lung involvement.

Disease severity was defined as mild (normal O_2_ saturation and normal chest X-ray), medium (radiological evidence of COVID-19 pneumonia), or severe, based on at least one of the following criteria: peripheral oxygen saturation (S_p_O_2_) ≤ 93% in ambient air, respiratory rate (RR) > 30/minute, arterial oxygenation partial pressure to fractional inspired oxygen ratio (P_a_O_2_/F_i_O_2_ ratio) < 300, or lung infiltrates > 50% of lung parenchyma).

Evidence for pulmonary bacterial infection was suggestive symptoms (e.g., fever, productive cough) or alveolar consolidation on chest CT, with or without positive microbiology. Given the fact that corticosteroids cause false leucocytosis with neutrophilia, and that COVID-19 induces a systemic inflammatory response, we considered that these biological markers had a lower contribution when assessing for bacterial infection. Isolated fever, in the absence of other clinical symptoms, with a procalcitonin value within normal range, was not considered sufficient reason to initiate antibiotic treatment.

Confirmed bacterial respiratory infection was diagnosed based on positive cultures of respiratory pathogens isolated from good-quality sputum (>25 polymorphonuclear leukocytes and <10 epithelial cells). Urinary tract infections (UTIs) were diagnosed based on symptoms and a positive urine culture (≥10^3^ UFC/mL). Bloodstream-invasive infections were defined as the growth of non-skin flora commensal on one or more blood cultures. For skin colonisers, we considered at least two positive blood cultures from different sets as microbiologically significant, unless the patient had high clinical suspicion and a predisposing condition (e.g., central venous catheter).

During their stay in the hospital, complete blood counts, inflammation markers, and d-dimer values were documented daily. We used the admission values, those prior to antibiotic administration (for patients who received antibiotics), and from the day with the highest CRP value (for patients who did not receive antibiotics).

The antimicrobial therapy was considered as appropriate for those who had reasons for pulmonary, urinary, or bloodstream infection, as previously described. Patients who received oral vancomycin for *Clostridioides difficile* colitis were included in the non-antibiotic group, even though they received empiric antibiotics before admission.

### 2.4. Sample Size

From the previous COVID-19 wave, in our hospital we had found a mortality of 10%, and a rate of antibiotic prescription of about 50%. For an alpha-error of 5%, a power of 80%, and an estimated 50% reduction in mortality in the antimicrobial therapy group, we calculated a sample size of 870 patients [10].

### 2.5. Data Analysis

Demographic, clinical, laboratory, biological, and imaging data of the enrolled patients were descriptively analysed. Continuous and categorical variables were presented as medians (min, max) and absolute numbers (percentage), respectively. A *p*-value of <0.05 was considered statistically significant. In the multivariable model, we introduced the variables associated with death with a *p*-value ≤ 0.10, and the regression was ruled stepwise forward because we had a relatively small number of patients who suffered that outcome. We analysed the collected data using the Statistical Package for Social Sciences (SPSS version 20, IBM Corp., Armonk, NY, USA) and Microsoft Excel 2018 (Microsoft Corporation, Redmond, WA, USA). The relative risk was calculated with EBM calculator (https://ebm-tools.knowledgetranslation.net/calculator, accessed on 3 March 2022) [11].

## 3. Results

A total of 700 patients was admitted into the six centres during the study period (January–May 2021). After eligibility assessment, 553 hospitalised COVID-19 patients were included in the study (Figure 1). The cohort had a median age of 67 (18–94) years, and 51% (283/553) of the enrolled patients were males. In terms of severity, 8% (44/553) of the patients had mild, 54% (297/553) moderate, and 38% (212/553) severe COVID-19. Three percent (14/553) of the patients were diagnosed with pulmonary thromboembolism (PE).

Treatment for COVID-19 included corticotherapy in 78% (431/553), anti-IL-6 tocilizumab in 6% (34/553), anti-IL-1-anakinra in 16% (89/553), and antivirals, including remdesivir in 22.4% (124/553), favipiravir in 15.6% (86/553) and umifenovir in 0.5% (3/553).

### 3.1. Bacteriological Results

Bacteriological tests requested by the attending physicians were performed in 57% (178/311) of the patients prior to ATB administration. After the microbiological screening, a total of 95 patients was diagnosed with bacterial infections. Positive results with clinical significance consisted of sputum examination (38 positive), urine cultures (31 positive), and blood cultures (39 positive). Based on the clinical and laboratory features, the organisms isolated were considered true infections, as opposed to colonisations. Ten percent (17/178) of them also had a fungal infection. Seven percent (12/178) of the screened patients had more than one positive bacteriological sample: sputum and urine culture (6/12), sputum and blood culture (2/12), urine and blood culture (3/12), or all three samples positive (1/12). In addition, 14 sputum and 2 urine cultures identified co-infections.

The detailed epidemiology of microbiological documented associated infections is presented in Table 1.

### 3.2. Antibiotic Prescription

A total of 58% (311/553) of the patients received antibiotic treatment during hospitalisation.

The patients who received ATBs were older, had a higher Charlson comorbidity index, and had more severe COVID-19. The median time for antibiotics prescription was 7 (min 0, max 78) days from symptom onset and 0 (min 0, max 24) days from admission. In total, 62% (193/311) of the patients received just one antibiotic during hospitalisation. The median number of antibiotics administered per patient was one (min 1, max 7). The most common classes of antibiotics prescribed were β-lactams, followed by fluoroquinolones and oxazolidinone (linezolid). When used in combination, carbapenems and oxazolidinone (linezolid) or vancomycin was the preferred regimen.

Antibiotics were administered according to antibiogram in 25% (77/311) of the patients. Empiric and antibiogram-guided antibiotic classes prescribed during hospitalisation are presented in Table 2.

Death was the outcome in 48 patients—39 from the ATBs group and 9 from the non-ATBs group. No difference was observed between the two groups in terms of median length of hospitalisation—13 days (min, max: 2–50) in the patients who survived vs. 12 days (min, max: 3–48) in the patients who died (*p* = 0.8).

### 3.3. Factors Associated with Patients’ Outcomes

The distribution of the variables in the deceased and surviving patients is presented in Table 3, as well as in the Appendix A (Appendix A).

In our study, the variables associated with death were older age, higher Charlson comorbidity index, COVID-19 severity, lower O_2_ saturation, higher inflammation markers (e.g., CRP, ferritin), and leucocyte and NLR values, as well as lower lymphocyte count.

The Kaplan–Meier curves for the whole group and subgroups (ATB and non-ATB) are presented in Figure 2.

Overall, the patients who received antibiotics during hospitalisation had a threefold higher mortality rate (12.5% vs. 3.7%, RR = 3.37, *p* < 0.001). In a subgroup analysis, among the patients without reasons for prescription, those who received antibiotics had a sixfold greater risk of death, while among those with reasons, there was no difference regarding the outcome—both in bivariate analysis (Table 3 and Appendix A), and after adjusting for the other prognostic variables (Table 4).

In the subgroup of patients with reasons for antibiotic prescription, only D-dimers (highest value) and S_p_O_2_ at admission were independent predictors of death (Table 5), while antibacterial therapy remained statistically associated with death even after adjustment among the patients without reasons for antibiotic treatment (Table 6).

The patients who died had a higher median duration from onset of COVID-19 symptoms to administration of antibiotics (11 vs. 8, *p* = 0.023), and received a median of two antibiotics during hospitalisation (min, max: 1–5).

With an AUROC value of 0.813 (0.744, 0.882), the highest D-dimer values documented during hospitalisation represented the best predictors of death. Moreover, the median value for D-dimers in the deceased group was four times higher compared with the patients who survived. Since there were no significant differences concerning the other laboratory variables between the values at admission and those during hospitalisation, the values at admission could be used.

A total of 48/91 tests for *C. difficile* came back positive; most of them were related to antibiotic prescription either during hospitalisation (39/48) or prior to admission (3/48).

## 4. Discussion

In our study, antibiotic administration did not decrease the mortality rate of patients with moderate and severe COVID-19, but it was associated with higher mortality in the subgroup of patients without documented bacterial infection. Previous studies have described the rates of antibiotic administration during the first and subsequent waves and the appropriateness of their prescription [12], but only a few of them assessed their impact on a hard outcome (death). As expected, in our study, the patients who died were older, had more comorbidities, and had more severe forms of COVID-19.

During the first waves of the pandemic, there was a high rate of antibiotic prescription—almost 70–80% of the hospitalised patients received antibiotics [13,14]. Azithromycin, ceftriaxone, and broad-spectrum antibiotics used to treat community-acquired pneumonia were the most commonly prescribed antibiotic regimens during the pandemic, probably due to the concern of bacterial co-infection [15,16]. Initially, azithromycin was also prescribed due to its supposed immunomodulatory and antiviral action, but later studies have shown no additional effect on survival or clinical outcomes compared to the standard of care [17,18,19]. In addition, regarding its antimicrobial effects, in Romania, as rates of macrolide resistance of *S. pneumoniae* are high, azithromycin is of little value in treating potential community bacterial co- or superinfections. During the pandemic, it was believed that prescribing antibiotics at large scale—especially in moderate and severe COVID-19 patients—might in fact have diminished the rate of bacterial infections [20], and probably helped save thousands of lives [21].

However, no more than 14.3% of hospitalised COVID-19 patients had a bacterial infection [22,23]. Seven percent (38/553) of our patients had an identified respiratory pathogen by positive sputum culture. As shown in previous studies, the rate of non-respiratory bacterial infections was even lower—only 0.8% of the patients had venous-catheter-related bacteraemia or urinary tract infections, and even fewer had intra-abdominal (0.3%) or skin and soft tissue (0.1%) infections [12]. In our study, the rate of non-respiratory bacterial infections was 11% (61/553), but since the patients enrolled had a relatively long stay in the hospital, some of these were hospital-acquired. In order to prevent overprescription, Giannella et al. proposed a predictive model using leucocytes, procalcitonin, and the Charlson index to quantify the risk of bacterial co-infection in hospitalised COVID-19 patients [24].

As expected, in our study, the patients who received antibiotics were those with more severe COVID-19; therefore, there was a selection bias that could have obscured any potential benefits of antibiotic therapy. Although we measured and adjusted for the variables proven until now to be correlated with death [25,26], it is possible that other unassessed variables might have been involved [27]. A severe evolution of the viral infection, unrelated to the bacterial superinfection, could also explain these results.

Pre-COVID-19 research showed that even when prescribed appropriately, antibiotics could not reverse the risk of death in the presence of other patient-related factors such as older age, neoplastic disease, non-ambulatory state, orientation disturbance, hypoalbuminaemia, tachycardia, tachypnoea, and hypoxemia—factors also frequently encountered in severe and moderate COVID-19 disease [28].

Due to the extensive bacterial screening, we consider the involvement of coexisting unidentified infectious agents or bacterial resistance less probable, and this was one of the reasons that we did not include patients previously admitted to the ICU. As expected, studies have shown that when adding hospitals to the community-acquired infections, the bacteraemia rate rises to 3.2–5.6% [29,30].

The main limitation of this study is its observational design. We adjusted for the factors known to be related to survival in order to address this issue.

When this project started, it was the first multicentre prospective study that aimed to evaluate the impact of antibiotic prescription on a hard outcome (death/survival). To date, three studies with a similar aim have been published, but two of them were retrospective and monocentric [31,32], and the third, although it had a large sample size, took into account only gender, age, and comorbidities as confounders [33]. However, their results are consistent with ours.

Overall, we showed that antibiotics, when prescribed in the absence of clear reasons suggestive of bacterial infection, were associated with higher mortality among patients with mild and severe COVID-19. In addition, our results could apply in the post-COVID-19 era as well, to patients with viral infections and/or immunosuppressive treatment, in whom unnecessary antibiotics might not positively influence the outcome.

## Figures and Tables

**Figure 1 jpm-12-00877-f001:**
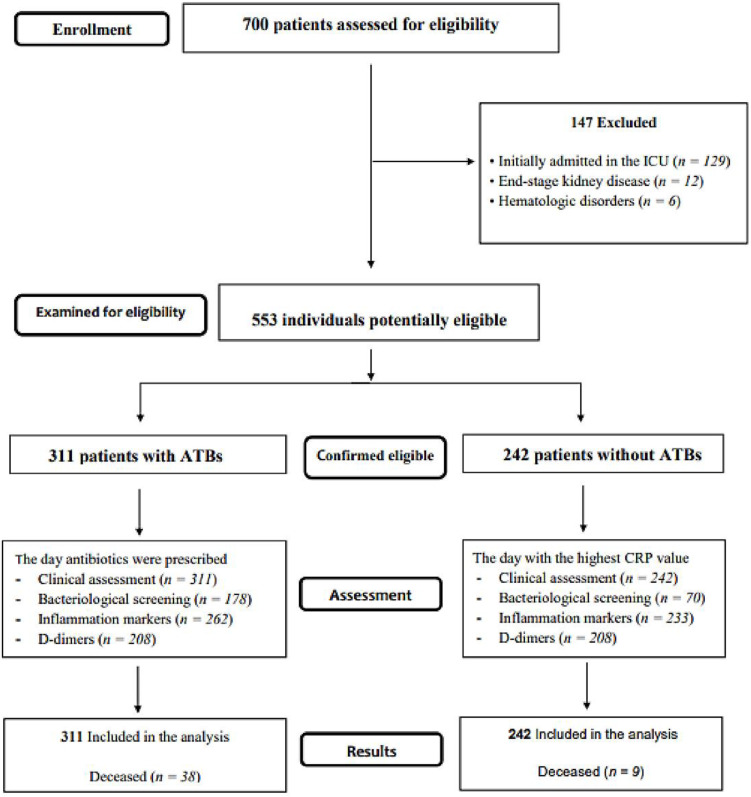
The flow diagram of the study describing the methods: participants’ recruitment, inclusion, exclusion, and outcomes.

**Figure 2 jpm-12-00877-f002:**
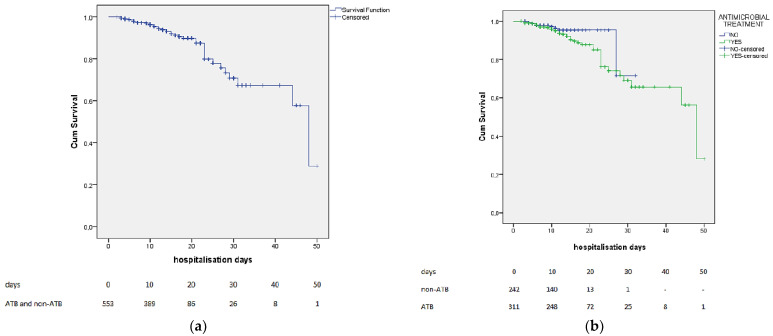
The Kaplan–Meier curves for the whole group (**a**) and subgroups (ATB and non-ATB) (**b**).

**Table 1 jpm-12-00877-t001:** Epidemiology of microbiological documented associated infections.

Aetiological Agents	Microbiological Sample *			Cases **, N
	Sputum	Urine	Blood	
**Gram-positive bacteria**				33
*S. aureus*				11
*Coagulase-negative staphylococci*				8
*Streptococcus* spp.				5
*Streptococcus pneumoniae*				5
*Enterococcus* spp.				4
**Fungi**				17
*Candida* spp.				16
*Aspergillus* spp.				1
**Gram-negative bacteria**				75
*K. pneumoniae*				26
*E. coli*				25
*P. aeruginosa*				4
*H. influenzae*				4
*A. baumannii*				3
*S. maltophilia*				3
*Serratia* spp.				2
*Hafnia paralvei*				2
*Sphingomonas paucimobilis*				2
*K. oxytoca*				1
*Enterobacter cloacae*				1
*P. fluorescens*				1
*Proteus mirabilis*				1

* The microbiological samples are marked by colour (blue for positive sputum samples, green for positive urine samples and red for positive blood samples), ** Multiple positive samples were identified per patient, and multiple pathological agents were identified per sample.

**Table 2 jpm-12-00877-t002:** Antibiotic prescription.

Antibiotic Classes	Patients *, N
**B-lactam drugs**	
Penicillins +/− BLI	85
Cephalosporins +/− BLI	163
Carbapenems	79
**Fluoroquinolones**	
Ciprofloxacin	14
Levofloxacin	26
Moxifloxacin	11
**Oxazolidinones** (linezolid)	27
**Cyclines**	
Doxycycline	13
Tigecycline	10
**Glycopeptides**	
Vancomycin	17
Teicoplanin	2
**Aminoglycosides**	
Gentamicin	4
Amikacin	6
**Macrolides** (clarithromycin)	3
**Others**	
TMP/SMX	23
Polymyxin E (colistin)	11
Metronidazole	11
Fosfomycin	4
Clindamycin	2
Chloramphenicol	1
Nitrofurantoin	1

* More than one antibiotic was administered in 38% of the patients.

**Table 3 jpm-12-00877-t003:** The distribution of the variables in the survivors vs. non-survivors groups.

Variable	Survivors, N = 505	Non-Survivors, N = 48	AUROC (CI 95%) * or RR (CI 95%) **	*p*-Value	Missing
Antibiotics during hospitalisation, N (%)	272 (53.9)	39 (81.2)	3.37 (1.7–6.8)	**<0.001**	
Prescription reason: yes, N (%)	223 (82)	30 (77)	2 (0.8–4.9)	0.124	
Prescription reason: no, N (%)	49 (18)	9 (23)	6.1 (1.9–19.1)	**0.001**	
Gender, female, N (%)	250 (49.5)	20 (41.7)	1.3 (0.8–2.3)	0.375	
Age, median (min, max)	67 (18–94)	74 (50–91)	0.642 (0.567–0.716)	**0.001**	
Charlson comorbidity index, median (min, max)	4 (0–12)	4.5 (1–11)	0.645 (0.567–0.724)	**0.001**	
COVID-19 severity (ordinal variable)				**<0.001**	
Mild	41 (8.1)	3 (6.2)			
Moderate	290 (57.4)	7 (14.6)			
Severe	174 (34.5)	38 (79.2)			
Pulmonary involvement ^$^, % (min, max)	40 (0–95)	70 (0–95)	0.662 (0.526–0.797)	**0.007**	
Pulmonary embolism, N (%)	13 (2.6)	1 (2.1)	0.8 (0.1–5.5)	1	
S_p_O_2__admission, median (min, max)	93 (60–99)	89.5 (58–99)	0.610 (0.518–0.701)	**0.012**	4
S_p_O_2__at ATB prescription, median (min, max)	93 (53, 99)	86.5 (56, 99)	0.660 (0.555–0.764)	**0.001**	66
Positive microbiology, N (%)	81 (29.8)	14 (35.9)		0.460	
Appropriate ATB, N (%)	74 (27.2)	3 (7.7)	0.23 (0.08–0.7)	**0.009**	
Corticotherapy, N (%)	395 (78.2)	36 (75)	0.85 (0.46–1.6)	0.588	
Tocilizumab, N (%)	28 (5.5)	6 (12.5)	2.2 (0.99–4.76)	0.105	
Anakinra, N (%)	73 (14.5)	16 (33.3)	2.6 (1.5–4.5)	**0.001**	
Antiviral	11 (22.9)	179 (35.4)		0.111	
CRP at admission, median (min, max)	55.7 (0.2–397.6)	80.2 (1.95–390.6)	0.583 (0.492–0.673)	0.060	9
CRP at ATB prescription, median (min, max)	60.4 (0.2–385.3)	86.5 (7.7–390.6)	0.616 (0.536–0.697)	**0.011**	58
IL-6 at admission, median (min, max)	30.2 (1–1406)	58.2 (7–656)	0.640 (0.536–0.743)	**0.016**	242
Leukocytes at admission, median (min, max)	7200 (1060–40,930)	8265 (2570–23,510)	0.591 (0.502–0.681)	**0.037**	9
Leukocytes at ATB prescription, median (min, max)	7670 (1060–29,760)	10180 (2570–28,570)	0.618 (0.527–0.710)	**0.009**	62
Neutrophils at admission, median (min, max)	5440 (650–36,790)	7135 (1120–20,410)	0.624 (0.539–0.709)	**0.005**	10
Neutrophils at ATB prescription, median (min, max)	6000 (650–24,690)	7810 (1800–26,400)	0.652 (0.567–0.738)	**0.001**	62
Lymphocytes at admission, median (min, max)	1000 (160–6470)	880 (150–5930)	0.606 (0.523–0.690)	**0.015**	10
Lymphocytes at ATB prescription, median (min, max)	1050 (160–4100)	850 (150–5930)	0.629 (0.541–0.717)	**0.005**	63
NLR at admission, median (min, max)	5.26 (0.29–60.8)	8.66 (0.69–84)	0.678 (0.604–0.753)	**<0.001**	11
NLR at ATB prescription, median (min, max)	5.42 (0.45–56)	9.13 (1.18–86.33)	0.698 (0.620–0.776)	**<0.001**	63
D-dimers at admission, median (min, max)	0.8 (0.1–15.4)	1.15 (0.1–7.2)	0.574 (0.481–0.667)	0.109	78
D-dimers highest value, median (min, max)	1.1 (0.1–20)	4.4 (0.5–20)	0.813 (0.744–0.882)	**<0.001**	74
D-dimers at ATB prescription, median (min, max)	0.7 (0.1–20)	1.9 (0.2–20)	0.677 (0.580–0.774)	**<0.001**	137
Urea at admission, median (min, max)	41 (11–189)	60.3 (7–129)	0.669 (0.591–0.748)	**<0.001**	17

AUROC—area under the receiver operating characteristic curve, CI—confidence interval, RR—relative risk, ATB—antibiotic, S_p_O_2_—oxygen saturation levels, CRP—C-reactive protein, IL-6—interleukin-6, NLR—neutrophil-to-lymphocyte ratio. ^$^ Assessed in only one centre. Statistically significant values are marked in bold. We calculated * AUROC for quantitative variables and ** RR for qualitative dichotomous variables.

**Table 4 jpm-12-00877-t004:** The association of antibiotic treatments with death after adjustment for the prognostic factors (logistic regression).

**Variable**	**Coefficient**	** *p* ** **-Value**	**Odds Ratio (95% CI)**
Age (years)	0.044	**0.049**	1.05 (1.0–1.09)
Form of disease *		**0.034**	
Moderate COVID-19	−0.864	0.482	0.4 (0.08–2.1)
Severe COVID-19	0.663	0.585	1.9 (0.18–20.9)
D-dimers (highest value)	0.250	**<0.001**	1.3 (1.15–1.43)
**Antibiotics (yes/no)**	**0.846**	**0.140**	**2.33 (0.76–7.17)**

CI—confidence interval. Statistically significant values are marked in bold. * Compared with mild COVID-19.

**Table 5 jpm-12-00877-t005:** The association of antibiotic treatment with death in the patients with reasons for prescription, after the adjustment for the prognostic factors (logistic regression).

Variable	Coefficient	*p*	Odds Ratio (95% CI)
D-dimers (highest value)	0.340	**<0.001**	1.41 (1.21–1.63)
S_p_O_2_ (admission)	−0.086	0.014	0.92 (0.86–0.98)
**Antibiotics (yes/no)**	**0.112**	**0.145**	**3.04 (0.68–13.55)**

S_p_O_2_—oxygen saturation levels, CI—confidence interval. Statistically significant values are marked in bold.

**Table 6 jpm-12-00877-t006:** The association of antibiotic treatment with death in the patients without reasons for prescription, after the adjustment for the prognostic factors (logistic regression).

Variable	Coefficient	*p*	Odds Ratio (95% CI)
Form of disease *		**0.038**	
Moderate COVID-19	−0.583	**0.728**	0.56 (0.02–14.9)
Severe COVID-19	1.586	**0.326**	4.9 (0.2–115.3)
D-dimers (highest value)	0.211	**0.009**	1.2 (1.05–1.44)
NLR (admission)	0.140	0.017	1.15 (1.02–1.3)
**Antibiotics (yes/no)**	**2.33**	**0.005**	**10.3 (2.0–52)**

CI—confidence interval, NLR—neutrophil-to-lymphocyte ratio. * Compared with mild COVID-19. Statistically significant values are marked in bold.

## Data Availability

The data presented in this study are available on request from the corresponding author.

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
