# Peer review of "Antibiotic Prescription and In-Hospital Mortality in COVID-19: A Prospective Multicentre Cohort Study"

_jpm, 2022, doi:10.3390/jpm12060877_

Round 1

Reviewer 1 Report

Pinte et al. report on antibiotic prescription and in-hospital mortality in COVID-19. The patient cohort is adequately powered. The manuscript is written in appropriate language, but requires more attention to detail 

Overall survival data (Kaplan-Meier analyses) should be provided for the whole group and subgroups, such as Antibiotics yes/no Table 3 is too lengthy, please move approximately 50% of the values to a supplementary table Table 3: The percentages are misleading and/or not carefully written down: e.g., prescription reason 223/272 = 82%. prescription reason 30/39 = 77%.  Table 4: "Form of disease" seems unclear, please explain/replace with a clear decription The discussion it too short and contains only a small number of citations. Given the plethora of COVID-19 publications, literature review should be performed more extensively Did the authors check for fever as reason for antibiotic coverage? It was only mentioned once.  The claim that antibiotics prescription in patients with COVID-19 without a clear reason is harmful is way too strong. The logistic regression presented here relies on 9 patients that died while receiving antibiotics with no obvious reasons.   MINOR:
No need for commata in title, please rephrase: "Antibiotic Prescription and In-Hospital Mortality in COVID-19: a Prospective Multicenter Cohort Study " Please use consistent abbreviations throughout the manuscript Please check for correct terms, e.g. "antibiotic prescribing" - ABT prescription "stafilococci" - staphilococci Please correct "physicians had been faced the difficult task to avoid overprescription" - physicians were challenged with the risk of overprescription

Author Response

Thank you for the assessment done on our manuscript and for the opportunity to revise our work. We’ve carefully read the comments and suggestions and prepared an improved version of the manuscript accordingly. Below please find attached a point-by-point response to the issues you raised and the revised document with changes highlighted throughout the manuscript.

Response to Reviewer 1 Comments

Thank you for the assessment done on our manuscript and for the opportunity to revise our work. We’ve carefully read the comments and suggestions and prepared an improved version of the manuscript accordingly. Below please find attached a point-by-point response to the issues you raised and the revised document with changes highlighted throughout the manuscript.

Pinte et al. report on antibiotic prescription and in-hospital mortality in COVID-19. The patient cohort is adequately powered. The manuscript is written in appropriate language, but requires more attention to detail

  1. Overall survival data (Kaplan-Meier analyses) should be provided for the whole group and subgroups, such as Antibiotics yes/no

The Kaplan–Meier curves for the whole group and subgroups (ATB and non-ATB) were included in the manuscript (Figure 2).

  1. Table 3 is too lengthy, please move approximately 50% of the values to a supplementary table

The table was modified according to the requirements and the remaining data was added in the supplemen (Table 1S).

  1. Table 3: The percentages are misleading and/or not carefully written down: e.g., prescription reason 223/272 = 82%. prescription reason 30/39 = 77%.

We've corrected the percentages.

  1. Table 4: "Form of disease" seems unclear, please explain/replace with a clear description

The meaning was added in the tables’ legends.

  1. The discussion it too short and contains only a small number of citations. Given the plethora of COVID-19 publications, literature review should be performed more extensively

We added the following paragraphs in the discussion section: ”Azithromycin, ceftriaxone and broad spectrum antibiotics used to treat community acquired pneumonia were the most commonly prescribed antibiotics during the pandemic, probably due to the concern of bacterial co-infection [15, 16].

Initially, azithromycin was also prescribed due to its supposed immunomodulatory and antiviral action, but later studies have shown no additional effect on survival or clinical outcomes compared to the standard of care [17-19].

In addition, regarding its antimicrobial effects, in Romania, as rates of macrolid resistance of S. pneumoniae are high, Azithromycin is of little value in treating potential community bacterial co- or superinfections.

During the pandemic, it was believed that prescribing antibiotics at large scale, especially in moderate and severe COVID-19 patients, might have in fact diminished the rate of bacterial infections  [20] and probably helped save thousands of lives [21].

As expected, studies have shown that when adding hospital to the community-acquired infections, the bacteraemia rate rises to 3.2-5.6% [29,30].

In order to prevent overprescription, Giannella et al. proposed a predictive model using leucocytes, procalcitonin and Charlson index to quantify the risk of bacterial co-infection in COVID-19 hospitalized patients [24]. ”

  1. Did the authors check for fever as reason for antibiotic coverage? It was only mentioned once.

We added in the Variables and data measurement section the following phrase: ”Isolated  fever, in the absence of other clinical symptoms or with procalcitonin value within normal range, was not considered sufficient reason to initiate antibiotic treatment.”

  1. The claim that antibiotics prescription in patients with COVID-19 without a clear reason is harmful is way too strong. The logistic regression presented here relies on 9 patients that died while receiving antibiotics with no obvious reasons.

We deleted the last sentence.

MINOR:

- No need for commata in title, please rephrase: "Antibiotic Prescription and In-Hospital Mortality in COVID-19: a Prospective Multicenter Cohort Study "

- Please use consistent abbreviations throughout the manuscript

- Please check for correct terms, e.g. "antibiotic prescribing" - ABT prescription "stafilococci" - staphilococci

- Please correct "physicians had been faced the difficult task to avoid overprescription" - physicians were challenged with the risk of overprescription

All the minor revision suggestions were addressed in the manuscript.

Reviewer 2 Report

Dear Authors,

Thank you for your submission. I enjoyed reading your manuscript. There are a few general suggestions for your consideration.

  1. There are minor capitalization errors throughout the paper. This error was most commonly seen with generic medication names and organism names such as line 118-119 with C. difficile spelled with a capital "D".
  2. The study aim was to assess the impact of antibiotic treatment on the outcome of hospitalized patients with COVID-19 as well as document the type of bacteria isolated and antibiotics prescribed.
    • In comparing the two group, a table that describes the demographics of both groups is not present.
    • It would be nice to have a table that compared the two groups similar to Table 3.
    • This study had dedicated a large portion of the manuscript to a sub-analysis to factors associated with survival.  Table 3 can be summarized into a smaller section. The remaining data can be supplied in a supplement. 
  3. Line 122-123, Sample size. Please provide further clarification on the calculated sample size. Typically a sample size is calculated to indicated a specific measurement. None is listed.
  4. Line 153, Table 1. While table 1 is interesting, it does not tell the reader were the organisms were isolated, nor whether the organisms were colonizers vs true infections.  Please indicate the number of co-infections as well.
    • The number of positive microbiology results in Line 148 has 38 sputum cultures, 31 urine cultures, and 39 blood cultures.  The total is 108.
    • Table 1 has 33 gram-positive organisms, 17 fungi, and 76 gram-negative organism. All totaling 126 isolates.  As mentioned previously clarity on the number of cultures that had multiple organisms.
    • Table 1,  The number of gram-negative bacteria cases are listed as 76, but when you add up all of the numbers below the total is 75. Please clarify.
  5. Table 2. Line 161. Please clarify whether the antibiotics listed in table 2 are empiric or directed therapy and which antibiotic combinations were most commonly used. The total number of patients on antibiotics is listed at 451, but the sample size was 311, which suggests that these are not unique values. Please provide more clarity on what is be represented. 
  6. Table 4, line 189 and table 6 line 195. Please include meaning in the legend  for form of the disease 2/1 and 3/1.  Line 92-101 was not clear on defining these terms. 
  7. Line 256-258. I disagree with the conclusion that antibiotics might increase the risk of death in patients without a clear justification. I don't think a clear link of causation of antibiotic use and mortality has been established.  A more appropriate conclusion of the study was the second to last sentence, line 254-256, " unnecessary antibiotics did not positive influence the outcome."  I recommend re-wording the last sentence or deleting it. 

Author Response

Thank you for the assessment done on our manuscript and for the opportunity to revise our work. We’ve carefully read the comments and suggestions and prepared an improved version of the manuscript accordingly. Below please find attached a point-by-point response to the issues you raised and the revised document with changes highlighted throughout the manuscript.

Response to Reviewer 2 Comments

Thank you for the assessment done on our manuscript and for the opportunity to revise our work. We’ve carefully read the comments and suggestions and prepared an improved version of the manuscript accordingly. Below please find attached a point-by-point response to the issues you raised and the revised document with changes highlighted throughout the manuscript.

Dear Authors,

Thank you for your submission. I enjoyed reading your manuscript. There are a few general suggestions for your consideration.

  1. There are minor capitalization errors throughout the paper. This error was most commonly seen with generic medication names and organism names such as line 118-119 with C. difficile spelled with a capital "D".

We made the changes in the text.

  1. The study aim was to assess the impact of antibiotic treatment on the outcome of hospitalized patients with COVID-19 as well as document the type of bacteria isolated and antibiotics prescribed.

In comparing the two group, a table that describes the demographics of both groups is not present.

It would be nice to have a table that compared the two groups similar to Table 3.

Unfortunately, the table that describes the demographics of both groups (ATB and non-ATB) is part of another article currently under revision at another journal. Since that article is a mixed-methods study which evaluates the factors associated with in-hospital antibiotic prescription during COVID-19 pandemic, and the doctors’ reasoning when deciding to administer antibacterial drugs, the table is better suited there.

Instead, we added a brief description of the two groups (ATB and non-ATB) in the section 3.2 Antibiotic prescription: ”The patients who received ATB were older, had higher Charlson Comorbidity Index and more severe COVID-19.”

  1. This study had dedicated a large portion of the manuscript to a sub-analysis to factors associated with survival. Table 3 can be summarized into a smaller section. The remaining data can be supplied in a supplement.

The table was modified according to the requirements and the remaining data was added in the supplement (Table 1S).

  1. Line 122-123, Sample size. Please provide further clarification on the calculated sample size. Typically a sample size is calculated to indicate a specific measurement. None is listed.

We changed in the text ”From the precedent COVID-19 wave, we had found a mortality of 10%, and a rate of antibiotic prescription of about 50% in our hospital. For an alpha-error of 5%, a power of 80%, and an estimated 50% reduction of mortality in the antimicrobial therapy group, we calculated a sample size of 870 patients.”

  1. Line 153, Table 1. While table 1 is interesting, it does not tell the reader were the organisms were isolated, nor whether the organisms were colonizers vs true infections. Please indicate the number of co-infections as well.

We added in the text. ” Based on the clinical and laboratory features, the organisms isolated were considered true infections, not colonizations.” The table was changed and the positive samples were mentioned. We added the number of co-infection in the text: ”In addition, 14  sputum and 2 urine cultures identified co-infections”

  1. The number of positive microbiology results in Line 148 has 38 sputum cultures, 31 urine cultures, and 39 blood cultures. The total is 108.

We changed the paragraph into: ”Bacteriological tests requested by the attending physicians were performed in 57% (178/311) of the patients prior ATB administration. After the bacteriological screening, a total of 95 patients were diagnosed with bacterial infections. Positive results with clinical significance consisted of sputum examination (38 positive), urine culture (31 positive) and blood cultures (39 positive). Based on the clinical and laboratory features the organisms isolated were considered true infections not colonizations.7% (12/178) of the screened patients had more than one positive bacteriological sample:  sputum and urine culture (6/12), sputum and blood culture (2/12), urine and blood culture (3/12) or all three samples were positive (1/12).”

  1. Table 1 has 33 gram-positive organisms, 17 fungi, and 76 gram-negative organism. All totaling 126 isolates. As mentioned previously clarity on the number of cultures that had multiple organisms.

We added in the tables legend: ”Multiple positive samples were identified/patient and multiple pathological agents were identified/sample.” We added the number of co-infections in the text: ”In addition, 14  sputum and 2 urine cultures identified co-infections”

  1. Table 1: The number of gram-negative bacteria cases are listed as 76, but when you add up all of the numbers below the total is 75. Please clarify.

The correct number of gram-negative bacteria is 75 - we made the change in the table.

  1. Table 2. Line 161. Please clarify whether the antibiotics listed in table 2 are empiric or directed therapy and which antibiotic combinations were most commonly used. The total number of patients on antibiotics is listed at 451, but the sample size was 311, which suggests that these are not unique values. Please provide more clarity on what is be represented.

The antibiotics listed in table 2 are both empiric or directed therapy and we added in the text: ”Empiric and antibiogram guided antibiotic classes prescribed during hospitalisation are presented in Table 2.”  In order to provide more clarity, we added in the text: ”When used in combinations, carbapenems and oxazolidinone (linezolid) or vancomycin, was the preferred regiment.” ”62% of the patients received just one ATB during hospitalisation.” We mentioned in the tables legend ”More than one antibiotic was administered in 38% of the patients.”

  1. Table 4, line 189 and table 6 line 195. Please include meaning in the legend for form of the disease 2/1 and 3/1. Line 92-101 was not clear on defining these terms.

The meaning was added in the table’s legends.

  1. Line 256-258. I disagree with the conclusion that antibiotics might increase the risk of death in patients without a clear justification. I don't think a clear link of causation of antibiotic use and mortality has been established. A more appropriate conclusion of the study was the second to last sentence, line 254-256, " unnecessary antibiotics did not positive influence the outcome." I recommend re-wording the last sentence or deleting it.

We deleted the last sentence.

Round 2

Reviewer 1 Report

Points were adequately addressed. Thank you